# Differences and commonalities in barriers and facilitators experienced by participants enrolled in an online behavioral weight management program: A qualitative comparison

Meigan Thomson[1]*, Anne Martin[1], Emily Long[1], Jennifer Logue[2], Sharon Anne Simpson[1]

1 MRC/CSO Social and Public Health Sciences Unit, University of Glasgow, Glasgow, United Kingdom,
2 Lancaster Medical School, University of Lancaster, Lancaster, United Kingdom

* meigan.thomson@glasgow.ac.uk

## Abstract

Online behavioral weight management programs offer a scalable solution to address overweight and obesity but often face high dropout rates and variable success. Understanding the barriers and facilitators experienced by participants who achieve weight loss targets versus those who do not is critical for optimizing programs. This study compared the barriers and facilitators reported by participants achieving ≥5% weight loss with those achieving <5%, using a social–ecological framework to capture influences across individual, interpersonal, and environmental levels. The framework was chosen to reflect the complex, interacting factors beyond the individual that shape behavior. Forty-eight participants completed semi-structured telephone interviews exploring factors affecting their weight loss journey. Interviews were analyzed using a thematic framework approach. Across both groups, key facilitators included willpower, knowledge, social support, and perceived safety in the local environment, with their absence acting as barriers. Food availability at home and in the workplace was also reported as a barrier due to temptation. Social roles could challenge behavior change, though perceiving themselves as role models provided motivation. Program-specific factors, such as group dynamics and difficulty using the platform, were barriers for both groups. Notable differences emerged between groups in responding to challenges. The ≥5% weight loss group proactively addressed stressors and sought solutions, while the <5% weight loss group reported greater difficulty overcoming barriers, more interpersonal stressors, and dissatisfaction with their weight targets. This is the first study to compare qualitative experiences across social–ecological domains between participants achieving ≥5% and <5% weight loss during program participation. Findings highlight the need to address environmental infrastructure, interpersonal skills, and communication of weight targets to improve program effectiveness. Future research should examine how these factors can identify individuals at risk of not achieving 5% weight loss and inform tailored interventions.

**Data availability statement:** The data generated and analysed during the current study will be archived in the UK Data Service ReShare repository: https://doi.org/10.5255/UKDA-SN-856651. Participants consented to data being shared for academic research and teaching purposes, therefore the data will have a restriction to ensure those accessing the data will be using it for this purpose.

**Funding:** This research is funded as part of an MRC PhD studentship (MC_ST_U18004). AM and SS were supported by UK Medical Research Council and Scottish Chief Scientist Office core funding as part of the MRC/CSO Social and Public Health Sciences Unit 'Complexity in Health Improvement' program (MC_UU_12017/14, MC_UU_00022/1 and SPHSU14, SPHSU16). EL was supported by MRC Skills Development Fellowship Award (MR/S015078/1) and the "Relationships and Health" program (MC_UU_00022/3 and SPHSU18). The funders had no role in study design, data collection and analysis, decision to publish or preparation of the manuscript.

**Competing interests:** JL is a now a fulltime employee of AstraZeneca with an honorary contract at Lancaster University. This research was completed prior to her employment at Astra Zeneca and they had no role in research. JL reported receiving personal fees for advisory boards from Novo Nordisk. This does not alter our adherence to PLOS ONE policies on sharing data and materials. The remaining authors have no conflicts to declare.

## Introduction

Globally the number of people living with obesity continues to increase [1]. This is concerning given the substantial threat obesity poses to both current and future mental and physical health [2,3]. Those living with overweight and obesity are at an increased risk of many conditions including cardiovascular disease, type 2 diabetes, several cancers, musculoskeletal conditions and depression. Recent evidence highlights that rising obesity rates carry significant health, social, and economic implications, contributing to growing pressures on individuals, communities, and healthcare systems [4,5]. As a result, there is growing public health interest to identify and improve interventions that can reach and benefit by large groups of people [6].

Online behavioral weight management programs (BWMPs) have the potential to reach large audiences and support them with weight loss. Such programs are delivered virtually and offer a combination of education on diet, physical activity (PA) and wellbeing, teaching behavior change techniques and providing social support to assist people living with overweight and obesity to improve their weight-related behaviors [7–9]. This combination of methods used in BWMPs have proven to harness better results and promote longer-term lifestyle and health improvement for those who complete the program compared to other weight management approaches [10–15].Within these programs, a goal of 5% weight loss is usually set as this has been shown to alleviate current and future ill-health [16]. Evidence indicates that OBWMPs can yield clinically meaningful results. For instance, a major commercial program found that more than 50% of its users achieved at least a 5% weight loss across six months [17]. Furthermore, within these programs,

Despite the success of BWMPs, they face significant challenges due to high attrition rates, which can range from 10% to 80% [18]. Dropping out of such programs is associated with poor weight loss outcomes and difficulty in maintaining any achieved weight loss compared to those who complete the program [19]. Even among those who manage to complete these programs, up to 50% still fail to achieve a weight loss of ≥5% [16] – this level of weight loss has been associated with improvements in present and future health [10,20]. Recent longitudinal evidence confirms that a 5% reduction is an optimal cut-off for meaningful cardiometabolic benefit over a 5-year period [21].

Consideration needs to be taken of factors internal and external to the program influencing outcomes. The social ecological model (SEM) provides a useful framework for conceptualizing how intrapersonal, interpersonal, environmental, and wider cultural and societal factors shape behavior [22]. The SEM considers how changes or interactions at one level may impact another level of the SEM. For example, changes in the physical environment (e.g., neighborhood walkability) may influence social interactions (e.g., ability to exercise with others) and subsequently physical activity behavior. The SEM posits these interactions may impact a person's behavioral reactions and promote or inhibit behavior change due to presenting different barriers and facilitators at these levels [22,23]. The SEM therefore suggests to effectively understand and develop interventions, we must understand how the levels interact and impact an individual's weight management behaviors. Given that participation in

BWMPs is integrated into individuals' daily lives, it is essential to consider how various social-ecological factors — from personal to environmental — shape the adoption of weight-related behavior changes.

Despite BWMPs typically being implemented within community settings, much of the research on barriers and facilitators to weight management has focused primarily on individual-level factors with only limited exploration of interpersonal influences. This approach often involves analyzing participants' experiences within the program (e.g., likes, dislikes, components they engage with), with some attention given to intrapersonal behavior and social support outside the program. Often wider influences as outlined in the SEM are neglected when exploring what helps or hinders weight management [18,24,25]. Studies have predominantly revealed that individuals who achieve a ≥ 5% weight loss tend to report higher levels of motivation, greater perceived control over their weight and behavior, increased self-efficacy and stronger social support [26–32]. Research on environmental influences underscores the significance of food availability, access to food and exercise facilities, the presence of green space, and the visibility of convenience foods in shaping obesity and weight-related behaviors [33–36]. Perceptions of the environment also influence weight-related behaviors. For example, Boehmer and colleagues found significant associations between obesity prevalence and perceptions that an area was predominantly residential, lacked points-of-interest (e.g., leisure facilities or green space) and had fewer or poorly maintained sidewalks [37].

Within the existing research, there is a notable absence of investigation and comparison regarding how these factors interact with an individual's weight loss efforts during their participation in BWMPs. Since BWMPs occur in community groups or online, understanding how levels of the SEM influence participation could inform program content and improve the implementation of the program's advice and guidance. Additionally, data collection typically occurs at the program's conclusion or during a follow-up phase, potentially overlooking essential nuances and meaningful barriers and facilitators experienced during participation. Furthermore, by comparing the barriers and facilitators encountered by individuals achieving ≥5% weight loss with those who do not, we can pinpoint commonalities and distinctive experiences in each group. Potentially this could assist with identifying individuals at risk of not reaching a ≥ 5% weight loss and additional provisions could be put in place to improve their chances of success. This insight would be valuable information to inform adaptations to interventions and enhance the overall effectiveness of BWMPs.

Therefore, this study aimed to compare barriers and facilitators experienced by ≥5% and <5% weight loss participants in an online BWMP. A qualitative approach was selected to explore their lived-experiences and understanding of how different factors impacted their weight-loss attempts. Following the SEM, this study aimed to understand how factors *external* to the program (i.e., intrapersonal, interpersonal, environment, and contextual) could influence weight loss during participation. Data were collected *during* participation to ensure factors which impact ability to make changes while using a program were identified. The study had the following objectives:

1. To gather qualitative experiences on barriers and facilitators faced by participants in a BWMP across social ecological domains

2. To identify commonalities in barriers and facilitators faced by those who achieve >5% weight loss and those who do not

3. To identify differences in barriers and facilitators experienced by those who achieve a > 5% weight loss and those who do not

## Materials and methods

### Study design

This study reports the qualitative findings from a wider mixed-methods doctoral thesis that examined barriers and facilitators of weight loss while participating in an online BWMP. The project collected data at three timepoints – baseline, mid-program and end of program. Survey data was collected at baseline and the end of the program, interview data

mid-way through the program and social network data at all timepoints. The wider findings are reported elsewhere [24]. Procedures and reporting of the qualitative data were conducted in accordance with the Consolidated Criteria for Reporting Qualitative Research guidance [38].

## Participant selection and sampling

Participants in this study were recruited from the Second Nature online BWMP in the UK. In the UK, participants can join Second Nature for free via referral from the National Health Service (NHS) or they can pay independently to join. Due to COVID-19 restrictions closing NHS weight management services, this study recruited those who paid to use the Second Nature platform.

**The Second Nature Program.** Second Nature is a 12-week online program that employs a combination of behavior change techniques, social support, and education to encourage modifications in weight-related behaviors. The program is structured such that participants are placed in an online group, guided by a dedicated "health coach" who leads them through the program [39]. The program provides a variety of recipes, self-monitoring and goal setting tools, and education on healthy behaviors each week such as diet, physical activity, stress management and sleep habits. Each week participants are required to update their measurements and where appropriate their goals. Education is informed by scientific research and healthcare guidelines and participants. The "Health coach" leads the education each week by sharing articles and facilitating discussions within the group chat. Within the group chat, members can speak to one another and share information which is monitored by the health coach who can comment to support problem solving or general discussion. Throughout the program participants can engage with materials as they wish and revisit previous articles and information.

**Recruitment.** The recruitment for the study took place from October to December 2020, with individuals who signed up during this period receiving an invitation email from Second Nature to participate in an interview. To be eligible, participants needed to be 18 years or older, have enrolled in the Second Nature program within two weeks of the recruitment period (this enabled baseline weights to be collected and interviews to be set up early/midway through the program), and have a BMI of 25 or above at the beginning of the program. Interested individuals who met the criteria contacted the lead researcher (MT) to learn more about the study, opt into the part of the study they were interested in (i.e., surveys, interviews, or both) and schedule an interview, if chosen.

Since it was uncertain how many participants would reach a ≥ 5% weight lost, a target sample size of 40–50 participants was set to ensure sufficient representation in both groups—those who met the 5% target and those who did not. Previous research indicates those meeting these targets are variable but typically just under half of participants will reach this level of weight loss [10], so the sample size was chosen to account for variability and to capture experiences from both outcomes. Participants who completed the interview received £20 in form of a shopping voucher as compensation for their time and contribution to the research.

## Data collection

Semi-structured telephone interviews were conducted with participants 4–8 weeks into the Second Nature program, with each interview lasting between 60–90 minutes. The interview schedule was designed to explore barriers and facilitators to weight loss within the framework of the SEM, encompassing environment, cultural, interpersonal, and intrapersonal factors. The schedule was developed using insights from the weight management literature and discussions with people with lived experience. The interviews took place during the COVID-19 pandemic, with additional questions related to COVID-19 included in the interview schedule. These covered how COVID-19 restrictions and policy impacted weight related behaviors and weight management. These results have been published elsewhere [40].

All interviews were recorded using an Olympus DS-9000 recording device and audio files were uploaded and stored securely. Verbatim transcriptions of the interviews were completed by a professional transcription company.

## Data analysis

Data were analyzed using NVivo 12 software [41]. A phenomenological and pragmatic approach were employed to explore how different constructs of the social-ecological model compare in the subjective experiences of barriers and facilitators to weight management between people who did or did not achieve a 5% weight loss during the program [42–44]. A 5% cut-off was selected since this has been shown to improve health and is a common target set by the program [8,17] Transcripts were analyzed using a thematic framework approach [45,46]. This approach was selected as it enables comparisons to be made between groups. The framework approach requires the researcher(s) to familiarize themselves with the data by reading transcripts and begin drafting a preliminary coding framework. The preliminary coding framework was then tested against 10% of the transcripts by two researchers (MT and either AM or SAS) independently, disagreements were resolved through discussion. This led to further refinement and finalizing of the coding framework. A mixture of deductive (i.e., structured by the social-ecological model categories) and inductive (i.e., data-led) coding was used. The coding framework was then applied to all transcripts. Transcripts were coded before participants final weights were collected so the researchers did not know their weight loss outcome while coding. All participants self-reported their body weights at baseline, time of interview, and at the end of the program. Baseline and end of program weights were used to determine the amount of weight participants lost. Once the amount of weight loss was calculated, participants were labelled as either successfully or unsuccessfully achieving a 5% weight loss after all transcripts were coded. This was to ensure amount of weight loss did not influence analysis and allow fair comparisons between groups. Following the labelling of participants, the data was charted into a framework matrix using NVIVo as per the framework approach. In the framework, each column represents a code and there was one row for participants reaching a 5% weight loss and one row for participants who did not reach the 5% weight loss. When a cell is selected it shows all the data coded to that code (column) to the participant group (row) and the researcher can summarize all the data in the cell. This allows for succinct summaries at each theme and enables comparison between the two groups to develop common and distinctive themes between groups.

## Ethical considerations

The study received ethical approval from the College of Social Sciences at the University of Glasgow (Approval number: 400190202). All participants received the information sheet, privacy notice, and consent form in advance and could ask questions via email or at the start of the interview. Written consent was obtained via email where feasible; otherwise, verbal consent was recorded at the outset of the interview and securely stored separately from the rest of the interview data. The researcher read each consent statement aloud, with participants confirming agreement and raising any questions. All audio files were uploaded and stored securely on University of Glasgow servers. Consent also covered data archiving for academic research and teaching.

## Reflexivity

The interviews were conducted solely by MT, who possesses a background in psychology and qualitative research methods. To ensure objectivity, the interview schedule was thoroughly reviewed and informed by input from the research team (AM, JL, EL, SAS), with expertise in the field, literature, and people with lived experience. People with lived experience were patient representatives identified via the Association for the Study of Obesity (who MT and AM were members of). Three patient representatives reviewed the interview schedules, and one took part in a pilot interview. These representatives provided feedback on the clarity of the questions and made suggestions for improvements. They recommended including questions on social norms and support within the household and workplace regarding dietary behaviors, as well as more detailed inquiries about their weight loss history. These suggestions were incorporated into the interview schedule. Furthermore, to reduce biases in data coding and interpretation, 10% of the transcripts were independently coded

by SAS and AM. In addition, MT remained blinded to participant outcomes during coding of all of the transcripts allowing for an unbiased comparison of themes between the groups. These measures strengthened the study's credibility and reduced potential biases throughout the research process.

## Results

### Participants

Forty-eight eligible participants completed the interview. Twenty-two of the 48 participants successfully achieved ≥5% weight loss during the program. Overall, the participants were mostly female (40/48) with a mean age of 49.1 (+/- 10.2) years and mean BMI of 31.6 (+/- 4.8) kg/m². A summary of participant characteristics is available in Table 1.

### Commonalities between groups

Participants who lost ≥5% weight over the course of the program shared many of the same barriers and facilitators to weight management than those who did not. These commonalities were found across the domains of the SEM. A summary of the identified themes present in both groups is shown in Table 2. The study identified one multi-domain theme which encompassed more than one aspect of the social ecological model (e.g., both intrapersonal and interpersonal factors). Policy and contextual themes focused on COVID-19 which has been reported elsewhere [40].

**Table 1. Demographic characteristics of participants.**

| | All (n = 48) | ≥5% *weight loss* (n = 22) | <5% weight loss (n = 26) |
|---|---|---|---|
| Gender (n (%), female) | 40 (83.33) | 18 (81.81) | 22 (84.62) |
| Age (mean, (range) years) | 49.09 (26-74) | 50.91 (34-63) | 47.48 (26-74) |
| Employment Status (n (%)) | | | |
| Working full-time (30 + hours per week) | 26 (54.17) | 14 (63.64) | 12 (46.15) |
| Working part-time (<30 hours per week) | 11 (22.92) | 5 (22.73) | 6 (23.08) |
| Unemployed and not seeking work | 3 (6.25) | 0 | 3 (11.54) |
| Retired | 3 (6.25) | 2 (9.09) | 1 (3.85) |
| Student | 2 (4.17) | 0 | 2 (7.69) |
| Carer | 1 (2.08) | 1 (4.55) | 0 |
| Furloughed | 1 (2.08) | 0 | 1 (3.85) |
| Missing data | 1 (2.08) | 0 | 1 (3.85) |
| Education (n (%)) | | | |
| High school | 3 (6.25) | 2 (9.09) | 1 (3.85) |
| Non-college/university qualifications | 15 (31.25) | 5 (22.73) | 10 (38.46) |
| Degree from college or university | 16 (33.33) | 9 (40.91) | 7 (26.92) |
| Higher degree (Master's. PhD) | 13 (27.08) | 6 (27.27) | 7 (26.92) |
| Other | 1 (2.08) | 0 | 1 (3.85) |
| Household Income (n (%)) | | | |
| £0-14 999 | 1 (2.08) | 0 | 1 (3.85) |
| £15 000–24 999 | 3 (6.25) | 0 | 3 (11.54) |
| £25 000–34 999 | 4 (8.33) | 2 (9.09) | 2 (7.69) |
| £35 000–51 999 | 11 (22.92) | 5 (22.72) | 6 (23.08) |
| £52 000–69 999 | 10 (20.83) | 5 (22.72) | 5 (19.23) |
| £70 000+ | 19 (39.58) | 10 (45.45) | 9 (34.62) |
| % weight loss from baseline to end (mean, (range)) | −4.64% (−17.00%-+7.20%) | −8.11% (−5.00%--17.00%) | −1.83% (−4.94%-+7.20%) |

**Table 2. Overview of barriers and facilitators experienced by both groups.**

| Social Ecological Domain | Themes |
|---|---|
| Intrapersonal | Willpower |
| | Knowledge |
| | Emotional Regulation |
| | Physiological Response |
| Interpersonal | Social Support |
| | Social Roles |
| Program-Specific | Group Relations |
| | Practical Issues |
| | Program Approach |
| Environmental (local) | Access to Obesogenic Amenities |
| | Safe Spaces for PA |
| Environmental (workplace) | Work Demands |
| | Lack of Facilities for Healthy Food |
| Multi-domain (i.e., themes which cover more than one aspect of the SEM) | Stigma |

**Intrapersonal commonalities.** Intrapersonal barriers and facilitators encompassed themes which related to the participants cognition, emotion, body, or personal circumstances. Overall, these themes acted as a facilitator when present and as a barrier when absent.

**Willpower:** Although participants were able to identify a range of barriers and facilitators across the SEM, there was consensus amongst all participants that success was ultimately attributed to their personal willpower:

"It's not rocket science really, it's the willpower. And it's the, I don't think that a diet doesn't succeed, it's the willpower that is just really difficult to find" (p29, <5%, female)

Participants described willpower as both feelings of motivation and control over their weight management.

**Knowledge:** The presence of knowledge was another theme highlighted by participants as contributing to weight management success. Participants discussed how learning new things about diet, PA, stress, and sleep enabled them to adopt new behaviors which supported their weight loss:

"I've learned that a lot of it is all about making good habits and things and not, and I didn't realise that was such a big part of it before." (p8, ≥5%, female)

Improving their understanding of weight loss trajectories also enabled them to manage any setbacks better and set more realistic goals in the future:

"You know, you plateau out, you stay with it and it will happen again, you'll start to lose weight again, just kind of hold fast to it." (p28, ≥5%, female)

Where participants struggled to understand advice, this acted as a barrier to weight management, as participants were unclear on how to implement changes:

"All the information that's out there is just so damn confusing, just all the: are carbs good, are carbs bad? Is fat good, is fat bad? The whole thing. The more you look into it, the more confusing it gets." (p25, <5%, female)

 

**Emotional regulation:** The ability of participants to manage their emotions significantly impacted their efforts to lose weight. Emotions related to setbacks in their weight management journey and personal circumstances played a crucial role. When participants struggled to control their emotions or cultivate positive feelings, it became a barrier and could lead to engaging in harmful behaviors like emotional eating or reduced motivation:

"When I feel good and dieting and moving around, I feel good emotionally. It's when something changes within my mood that I suddenly give up» (p7, <5% female)

On the other hand, those who could harness positive emotions in the face of negativity found it beneficial for their weight loss. They often used weight-regulatory behaviors like PA to regulate their emotions and enhance their well-being. Additionally, they employed reasoning strategies to handle setbacks and adopt a positive mindset for continued progress:

"If I had a hiccup, for example, if my wife and daughter say, okay, we fancy pizza tonight and I go, okay, I'll have a slice. Then I reset the next day and start again" (p44, <5%, male)

**Physiological response:** Participants described having a positive physiological response to dietary and PA changes as a facilitator of weight loss. These responses included improvements in physical appearance and how clothes fit, feeling satiated and improvements in physical health and energy levels. When participants had a negative reaction such as physical pain this could result in termination of changes. However, in some cases, physical pain could facilitate the continuation of a new behavior:

"I had awful headaches and everything for about the first four or five weeks, but I think I've got through that now, so I don't want to eat sugar, and then put myself into the position of having to go through that all again, you know." (p6, ≥5%, female)

**Interpersonal commonalities.** Interpersonal barriers and facilitators encompassed themes which related to social factors.

**Social support:** Participants in the study highlighted the significance of social support in facilitating weight-related behavior changes. This support encompassed both practical support, such as engaging in joint meal preparation or exercise, as well as emotional support, like providing empathy and being attentive listeners to their problems. Participants perceived social support as more supportive of weight loss when people they were close to (e.g., within the household) offered a combination of emotional and practical support:

"My partner's sat and he's read the handbook, he's looked at recipes. My youngest daughter who's still at home, she's looked at recipes, can we have this? You know, they are supportive... Supportive of me wanting to lose weight." (p29, <5%, female)

Both practical and emotional support played a crucial role in helping them overcome challenges and stay committed to their weight-related goals.

**Social roles:** Social roles influenced how well a participant could implement weight-related behavior changes. For some, their social roles presented challenges to changing their behaviors. For example, participants who identified as "feeders" and enjoyed preparing meals and hosting people to show they care found it challenging to change eating behaviors and how they socialized using food because of this identity. They struggled to consider alternative ways to show this form of care and did not want to prepare meals others would not like. While others felt their professional role (e.g., particularly those in a healthcare role) put added pressure on their weight management. For example, worrying about how weight

changes could impact how others respond to you and your relatability to people in your role could produce mixed motivations towards weight loss:

"…it feels a bit easier to talk about it with them if you are a bit overweight…you wonder whether that makes it less intimidating for the patient and less but then… I don't know…any doctors that are overweight I don't think" (p12, ≥5%, male)

**Program-specific commonalities.** Program-specific barriers and facilitators encompassed themes directly related to the program's environment, content, or social dynamics.

**Group Relations:** Group relations were identified by participants as both a potential barrier and a facilitator. When participants found it challenging to engage with the group or felt the communication demands were too intense, this could lead to disengagement:

"I even find that a little bit not anxiety provoking but I feel like I need to…I'm meant to put something into it, type something into it and, you know…or like chase something up, contribute to the group, but I just don't to be honest. And then I feel a bit guilty about not doing it." (p12, ≥5%, male)

Conversely, some participants viewed the group as a valuable means of connection and a meaningful source of support and advice, which helped sustain their commitment to the program during challenging periods:

"I've explained just like, oh, I'm overwhelmed with stuff and it's been good and I've found a few people, there've been two or three other people [in the group] with chronic illnesses and we've been…started chatting I guess about…chronic illness stuff and then also kind of grouped off sideways… extra bits, and exercises on chairs and some extra support." (p9, <5%, female)

"You feel supported and you feel like other people are also struggling with weight and trying to lose weight, and it gives you a greater sense of motivation." (p14, ≥5%, female)

Similarly, perceptions of the health coach influenced engagement. When the coach was seen as informative, encouraging, and responsive, this fostered greater engagement with both the materials and the wider group:

"And she just told me to focus on one or two things that you can just commit to for this week and just focus on those. And actually, that worked really well. So, I've done…did that and committed to, I think, three things actually. And absolutely did those and now I feel right back into it again." (p1, <5%, male)

However, when participants perceived the coach's communication as too "scripted" or "robotic," it led to disengagement:

"It didn't feel enough, and you know, it's the same again, you couldn't really build up a rapport with them." (p4, <5%, female)

**Practical Issues:** Practical challenges also emerged, including difficulties navigating the online platform, accessing resources, inputting data, and balancing the program with other life commitments:

"I think logging it is really important for me, but the planning, and planning and logging in two separate places, that was really annoying." (p4, <5%, female)

"I find it difficult that everything's on the app, the app I don't think is particularly great designed, and yeah, it's slightly difficult, because sometimes I feel like I'd rather do things on the computer screen, but the computer screen doesn't have the same…when you log in on the computer, it's not the same as the app, and it's difficult to find things." (p26, ≥5%, male)

**Program Approach:** Participants' views on the program's approach also influenced their engagement and trust in the advice provided. For many, the use of scientific articles, a mature tone, and NHS backing enhanced the credibility of the information and boosted motivation:

"It seems quite grown up and doesn't talk down to you, it's intelligent. And I like the fact that the articles are well written and clearly explained and without being too basic. There's some interesting scientific stuff." (p23, <5%, female)

Conversely, when some advice was perceived as difficult to believe, participants struggled to follow it:

"I disagree with some of the advice given and I would disagree with some of the psychology around habit forming and stuff like that. I think some of it's a little bit…it's not particularly based on good science, it's pop psychology." (p30, ≥5%, male)

Participants also noted that the program's holistic approach supported their success by encouraging them to consider broader lifestyle and wellbeing factors, leading to better outcomes:

"I think they're really good, because they focus on so many different aspects, like the time you have the healthy choices that you make, your food or your sleep patterns, how they influence your choices of food and things." (p14, ≥5%, female)

**Environmental commonalities.** Environmental themes related to how place could act as a barrier or facilitator to weight management. Largely, these were barriers and could be grouped as local or within the workplace.

**Access to obesogenic facilities:** Access to obesogenic facilities was reported as a barrier to weight management due to increased temptation. Food accessibility was reported as a barrier in the home especially where other household members did not change their dietary habits. Outside the home issues arose where participants lived within close proximity to retail and fast-food outlets, as well as having a lack of control over menu options and portion sizes at outlets:

"I think eating out is a problem. I think one of the main problems is portion size, eating out is always too much. Always too much." (p3, ≥5%, female)

Many participants also noted unhealthy options are usually more affordable or cost-effective than healthier options.

**Safe spaces for PA:** Participants noted the presence of green space as a facilitator for their weight management. Green space was described as encouraging PA due to the positive impacts on mental wellbeing. Barriers occurred when participants could not physically access green space due to unsuitable walking paths (e.g., uneven terrain, muddy, poor lighting), overcrowding, busy traffic, and knowledge of crime in the area:

"I do when it's daylight hours. If it's on a night, there is no street lighting there anyway, it's literally the edge of the river…it's really boggy, I mean, I got my shoes stuck the other day, trying to go that way" (p17, <5%, female)

**Lack of facilities for healthy food in the workplace:** A major barrier to weight management was lack of control over meal preparation and food choices in the workplace. Participants reported a lack of places to purchase healthy food and prepare food and having increased accessibility to unhealthy options:

"I would be buying things in the canteen and its quite big meals and not the healthiest, and also people bringing things in, you know, like for everybody, like sweets and cakes and, you know, that type of thing, that can make it difficult. That bit temptation" (p31, ≥5%, female)

**Multi-domain commonalities. Stigma:** Stigma emerged as a significant obstacle to weight management, described by participants across two domains. These included interpersonal and intrapersonal stigma. Negative comments from others and media depictions of obesity contributed to the stigma experienced. A participant even faced stigma from her child after he learned about obesity and healthy lifestyles in school:

"…there's actually a lot of discussion about unfit and overweight at school and it's not really had a positive…it seems like more stigma rather than less coming out of that discussion at school" (p9, <5%, female)

Moreover, intrapersonal stigma was evident, as participants expressed biases in their social interactions and used derogatory language (e.g., "fatty") when referring to themselves. Participants noted experiencing stigma could result in lower mood which could hinder weight behavior changes.

## Differences between groups

Across the SEM there were some key distinctions between those who achieved a ≥ *5% weight loss and those who did not.* Differences appeared in how groups perceived and reacted to barriers. An overview is provided in Table 3. Notably the differences show <5% weight loss participants reported distinct barriers and ≥*5% participants reported distinct facilitators.*

**Awareness of barriers.** Groups differed in their awareness of potential barriers to weight management. Those who achieved a ≥ 5% weight loss considered more barriers and facilitators across social ecological domains than those with <5% weight loss. They were also mindful of how different barriers could interact *with one another.* They particularly highlighted the role of negative media and public health messaging on weight management:

"I recognise obviously there are health benefits and they should put in, you know, the public health messages about weight loss, and stuff, but equally it also makes people who are overweight constantly feel shit, which often doesn't help to, you know, encourage people to lose weight because if you feel shit you often just need to eat bad stuff because it gives you some fleeting comfort." (p12, ≥5%, male)

**Motivational differences.** The primary motivation for weight loss among all groups was to enhance or preserve their health. However, in the ≥ 5% group an additional motivation was wanting to lose weight for the well-being of others. This included being able to actively engage with grandchildren, supporting their children by prolonging their life, and preventing loved ones from facing difficulties if they were to become unwell due to their weight:

"What's hit me is how badly my two sons have reacted to their father's death, and it's made me realise, I don't want to put them through that. I know eventually they will, but I don't want to do it in the next five or ten years, I want to be staying fit and be healthy." (p6, ≥5%, female)

Additionally, this group reported becoming more motivated by stressors and hardships. They found inspiration in reflecting on their past triumphs, empowering them to confront challenges related to their weight management.

**Reactions to setbacks, barriers and stressors.** As well as enhancing motivation, ≥ 5% weight loss group were more pragmatic when faced with barriers, setbacks, and stressors. They described developing solutions which could easily be implemented in the situation to support their weight loss:

**Table 3. Differences in reported barriers and facilitators between groups.**

| Theme | <5% group | ≥5 group |
|---|---|---|
| Awareness of barriers | Less aware of external barriers | Considered barriers and how they interacted more holistically *(facilitator)* |
| Motivational differences | No distinction | Motivated by other's wellbeing *(facilitator)* |
| | | Motivated by stressors *(facilitator)* |
| Reactions to setbacks, barriers & stressors | Difficult to navigate and felt less in control *(barrier)* | Sourced pragmatic solutions *(facilitator)* |
| Reactions to weight loss target | Disliked weight targets *(barrier)* | Proactive in learning risks of weight & benefits of weight loss *(facilitator)* |
| Interpersonal conflict | Negative comments or reactions from others about their weight loss *(barrier)* | No distinction |
| | Resistant to receiving support from others *(barrier)* | |
| Work patterns | Long commutes, shift work and long hours acted as barriers | No distinction |

"We're going to a Mexican, so I'm kind of hoping that I can…I was thinking of making my own keto fajitas and taking my own wraps and asking them to serve it with my own wraps, rather than their wraps." (p26, ≥5%, male)

Whereas <5% participants felt a lack of control when experiencing a setback or facing a barrier to their weight management:

"I just don't get why I can't get back into it. Whether it's because of the added stress of, you know, doing school runs or at weekends it's only just me and my son, so not being able to have that flexibility and trying to fit it all in during the week. I don't know, it's something I can't work out." (p20, <5%, female)

**Reactions to weight loss targets.** Groups had notable differences in how they reported reacting to their weight loss targets suggested by the program. The ≥5% groups actively engaged in learning about the risks associated with their weight and the benefits of weight loss:

"I was looking on the NHS website, you put your weight in and your height and it shows you your BMI and it tells you about the risks and then gives you options, suggestions, like you should try to speak to your doctor." (p14, ≥5%, female)

Conversely, the <5% group were more likely to find these targets unacceptable or believed that achieving them would lead to unhealthy outcomes. This group appeared less inclined to embrace the program's goals and were sceptical about their feasibility or potential health implications:

"I'm trying not to say big-boned but I am. I've just got very solid bones and when I had the photographs taken of my hip at the hospital the other day my hip bones go right out to the edge of me. So I'm never…I'm going to be a big skeleton with no flesh on if I keep going down to my BMI weight which I think is 10 and a half stone. I'd just look absolutely dreadful." (p11, <5%, female)

**Interpersonal conflict.** Participants who lost <5% of their weight reported interpersonal conflict as a major barrier to their weight management. This included experiencing negative comments or reactions to their weight loss attempts,

feeling rebellious towards social desirability to be thin, and being more resistant to receive support from people in their lives.

"He's nagged for most of the last 15-20 years: you need to lose weight, you need to lose weight, and each time I put dinner out, he will say, I hope you're cutting back. It's almost like the voice in the head, kind of thing. And I guess part of that might be why I still sneak food and hide food, so that he doesn't know it's there either." (p25, <5%, female)

**Work patterns.** Long commutes, working shifts or long hours was reported as a barrier to change in the <5% group. Participants reported this inhibited their ability to form a routine, incorporate PA into their daily lives and have time to cook fresh meals.

"I could be in the car for about six hours, like, a round trip and then could be out the car for five/six hours and you can't wear your smartwatches or anything like that in there… sometimes I try and plan my food for the day to take with me but if I'm not prepared then I end up just buying something" (p20, <5%, female)

## Discussion

This is the first study to compare qualitative barriers and facilitators reported by ≥5% and <5% weight loss groups. Importantly, it captures these factors during program participation rather than relying on follow-up data collection, which can be affected by hindsight bias. This approach strengthens the study by offering real-time insights into the challenges participants face and a more nuanced understanding of the factors influencing weight management as they occur. The research revealed that many of the barriers and facilitators, spanning the SEM, were common between groups, though some distinctive factors were identified. Suggestions for BWMPs based on the findings of this study are listed in Table 4.

**Table 4. Suggestions for behavioral weight management programs based on findings.**

| Finding from the study | Consideration for BWMPs |
| --- | --- |
| Willpower was highlighted by all participants as the deciding determinant of success | Support individuals to identify key motivations and problem solving for barriers to weight management |
| Confidence in levels of knowledge and trust in weight targets were reported as facilitators to change | Offer 1:1 sessions with participants to assess their understanding and offer tailored justifications for weight targets |
| Physiological responses to dietary and physical activity changes affected engagement | Offer 1:1 sessions with participants to assess any unwarranted physiological response to adapt advice |
| Social support was identified as an important contributor to success. <5% participants reported experiencing more interpersonal conflict related to their weight | Offer social support within programs and advice in harnessing support from their networks. Include social skill development and conflict resolution skills within programs. |
| Positive emotional regulation and reaction to setbacks were identified as facilitators | Include emotion regulation strategies, help participants to identify triggers and ways to relax, offer emotional support. |
| The built environment acted as a barrier to weight management through temptation accessibility and lack of spaces to prepare food or engage in PA. | Tailor program advice related to PA and diet to the local area. Offering healthy food options to prepare and physical activity that can be done at home. Consider options for participants who have long or unsociable work hours. |
| ≥5% were more aware of barriers/facilitators across domains and were able to source solutions to barriers | Support individuals to think more holistically about influential factors on their weight. Include problem solving strategies and techniques for them to incorporate into daily life |
| <5% participants identified more barriers | Include assessments within BWMP of barriers participants are facing. Offer further problem-solving support to those who identify more barriers. |

This research underscored the significance of various factors related to the individual, relationships with others, and factors in the surrounding environment. Particularly noteworthy is the unanimous emphasis among all participants on the pivotal role of willpower in shaping outcomes. Likewise, a qualitative synthesis of obstacles in weight management among individuals living with obesity identified the central importance of willpower as a critical determinant [26]. Although challenging to define, the concept of willpower may be understood as having higher levels of motivation and self-efficacy which supports weight management and behavioral changes [47–49]. Participants also highlighted the significance of knowledge, emotional regulation, and physiological responses in their weight management. Poor emotional regulation was linked to eating in response to negative emotions and difficulty in managing stressors. Emotional regulation and emotional eating are often reported as a barrier to weight management in weight management research both within and outside the context of COVID-19 [50–53]. While feeling satiated is recognized as important for weight management [54,55], this study revealed less-explored themes related to physiological reactions to changes, such as feeling more energetic or experiencing discomfort.

In the interpersonal domain, both groups emphasized the significance of social support and their social roles in shaping their capacity to make changes. The study highlighted that a combination of emotional and practical support plays a crucial role, aligning with existing literature [32,56]. A novel finding in this study is some participants faced challenges in implementing behavioral changes due to conflicts with their identities or perceived social roles. Research shows who we spend time with influences our weight-related health behaviors [57], this includes levels of support [32,58,59] and general influence around healthful behaviors [60,61].

Participants reported several challenges within the program. Difficulties in integrating into the group or establishing rapport with the health coach were noted as barriers to feeling part of the community. Conversely, when participants found the content and coach engaging, accessible, and credible, it served as a facilitator; however, challenges in understanding the material acted as an obstacle. This corresponds with results from a component analysis of online BWMPs, which found that credible information, engagement with a trusted specialist, and appropriately tailored incentives were significantly linked to successful weight loss. [62]. Practical issues also posed challenges, including navigating the online platform, accessing resources, inputting data, and balancing the program with other life commitments. Importantly, these program-specific barriers and facilitators were common across all participants, regardless of whether they achieved a 5% weight loss. This suggests that external factors beyond the online program may have had a more significant influence on weight-related behavioral changes. In the broader environment, participants identified several issues. These included a lack of visibility and control over restaurants and food outlets, insufficient safe spaces for physical activity, and limited access to and preparation of healthy foods in the workplace, which has also been found in the wider literature [63–65]. Hosting such interventions within the workplace has been efficacious in supporting positive weight management practices [65] and allows interventions to be tailored to the unique obstacles workers may face [66–68].

The theme of stigma was identified across multiple domains of the SEM. The evidence indicates that weight stigma is widespread and deeply rooted in society, significantly affecting the well-being of people living with obesity [69,70]. Stigma may influence motivation negatively by reducing self-esteem, increasing body image disturbance, and increasing levels of psychological distress [71]. However, Fogel and colleagues found stigma can be harnessed to foster motivation through becoming a role model to others. In a weight management program tailored for lesbian women, challenging stereotypes, and encouraging other lesbian women increased motivation to engage with the program [72].

Notable differences were found between the groups. A key defining element of the data was the < 5% weight loss group identified distinct barriers whereas the ≥ 5% weight loss group identified key facilitators. Zheng and colleagues similarly found that "good responders" (i.e., > 15% weight loss) reported fewer barriers to their weight management than those who lost <5%, suggesting actual or perceptions of barriers may differ between groups [73]. The reporting of barriers by participants could potentially serve as an indicator to identify individuals at risk of not achieving their targets during the program,

thereby enabling the provision of additional support. Future research and BWMPs should consider developing tools to assess the number of barriers experienced and develop thresholds for where further support may be needed.

Responses to challenges or obstacles also differed between groups. The ≥ 5% group considered a broader range of factors from the SEM when discussing barriers and facilitators. They demonstrated greater motivation, actively sought practical solutions to barriers, and took a proactive approach to understanding their weight and associated risks. Conversely, the < 5% group faced difficulties in dealing with obstacles and encountered more interpersonal barriers. This suggests that participants who lose <5% weight may find it fundamentally challenging to recognize the influence of broader factors on their weight management.

Moreover, our research discovered <5% group reported being more discontent with their weight loss targets suggested by the program. Interestingly, a recent prospective longitudinal study conducted by Wren and colleagues found participants who set larger weight loss targets were more successful and less likely to drop out of the Second Nature program [74]. These findings highlight the significance of weight loss targets in determining the success of participants. To better understand how to support participants in setting meaningful targets without leading to early disengagement, further research is required. Furthermore, the targets set by participants could serve as an indicator of whether additional support is needed.

The < 5% group reporting more barriers may be indicative of their readiness to change or differences between groups in their extrinsic/intrinsic motivation as posed by self-determination theory [75,76]. The focus on facilitators and managing potentially corrupting situations by the ≥ 5% group may indicate higher levels of intrinsic motivation, competence and autonomy as posed by the theory [47].

It is important to acknowledge the limitations of this study. Firstly, data collection took place during the COVID-19 pandemic in the UK, which introduced varying levels of social and environmental restrictions. The association of poor COVID-19 outcomes in people living with obesity was well-documented and publicized which may have influenced motivations to lose weight [77,78]. Furthermore, the closure of many social and environmental amenities and this may have biased the data towards intrapersonal themes. The study only identified policy-related themes that were linked to COVID-19 restrictions, which have been previously reported elsewhere [40]. As a result, the study lacks insights into non-COVID-19 related policies that can influence weight management, such as sugar tax or other relevant policies. This limitation suggests a need for further research to explore the impact of various policy measures on weight management outcomes beyond the scope of COVID-19 restrictions.

Moreover, the data mainly focuses on barriers and facilitators encountered by female participants with higher income and education levels. It is probable that male or non-binary individuals, as well as those with lower incomes or less formal education may experience distinct barriers and facilitators across various social ecological domains. It is also common for people living with obesity to experience a comorbid health condition, which could interact with how barriers and facilitators manifest. For example, a comorbid condition could affect motivation, access to resources, or the perceived cost-benefit of behavior change [79]. Additionally, examining obstacles related to resource availability and environmental infrastructure could provide valuable insights into the challenges faced by individuals in deprived areas and communities [80]. Additionally, it is important to note that the data were collected at a single timepoint during the early to mid-stages of the program which may have prevented the identification of barriers and facilitators that arise later in the program. Such factors are likely to evolve over time depending on individuals' weight-loss experiences. To better understand how barriers and facilitators change and how they relate to different weight-loss trajectories, future research should incorporate interviews at multiple timepoints.

Despite these limitations, the study offers valuable insights into barriers and facilitators experienced across different domains of the SEM. Although COVID-19 restrictions may have impacted the results, the identified themes are still relevant and applicable in a non-COVID-19 context and have been supported by other research. Nevertheless, future research should consider exploring environmental, policy, and societal/cultural barriers and facilitators more comprehensively.

## Conclusion

This is the first study to compare barriers and facilitators experienced across the social ecological model for participants attaining a ≥ 5% versus <5% weight loss. These findings highlight areas that programs can target to better support participants, particularly those likely to achieve less than a 5% weight loss. Key areas for considerations for programs include helping participants manage reactions to barriers, addressing social conflict and support, and providing tailored advice to where the program is situated. Future research should consider how to further differentiate those who reach weight loss goals with those who do not uncover further areas where support can be added across social ecological domains. Tracking barriers and facilitators across the participant journey could enable key milestones to be identified.

## Acknowledgments

The authors express their gratitude to all the participants who participated in the interviews. Additionally, they extend thanks to the Second Nature team for their support in recruiting participants for the study. However, it is important to note that Second Nature had no involvement or influence in the study's design, data analysis, or data interpretation.

## Author contributions

**Conceptualization:** Meigan Thomson, Anne Martin, Emily Long, Jennifer Logue, Sharon Anne Simpson.

**Data curation:** Meigan Thomson.

**Formal analysis:** Meigan Thomson, Anne Martin, Sharon Anne Simpson.

**Funding acquisition:** Jennifer Logue, Sharon Anne Simpson.

**Investigation:** Meigan Thomson.

**Methodology:** Meigan Thomson, Anne Martin, Jennifer Logue, Sharon Anne Simpson.

**Project administration:** Meigan Thomson.

**Resources:** Meigan Thomson, Jennifer Logue.

**Software:** Meigan Thomson.

**Supervision:** Anne Martin, Emily Long, Jennifer Logue, Sharon Anne Simpson.

**Writing – original draft:** Meigan Thomson.

**Writing – review & editing:** Meigan Thomson, Anne Martin, Emily Long, Jennifer Logue, Sharon Anne Simpson.

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
