## [Decision Letter · Decision Letter 0]

12 Oct 2025

Dear Dr. Thomson,

Thank you for submitting your manuscript to PLOS ONE. After careful consideration, we feel that it has merit but does not fully meet PLOS ONE’s publication criteria as it currently stands. Therefore, we invite you to submit a revised version of the manuscript that addresses the points raised during the review process.

If applicable, we recommend that you deposit your laboratory protocols in protocols.io to enhance the reproducibility of your results. Protocols.io assigns your protocol its own identifier (DOI) so that it can be cited independently in the future. For instructions see: https://journals.plos.org/plosone/s/submission-guidelines#loc-laboratory-protocols. Additionally, PLOS ONE offers an option for publishing peer-reviewed Lab Protocol articles, which describe protocols hosted on protocols.io. Read more information on sharing protocols at https://plos.org/protocols?utm_medium=editorial-email&utm_source=authorletters&utm_campaign=protocols .

We look forward to receiving your revised manuscript.

Kind regards,

Taiwo Opeyemi Aremu, MD, MPH, PhD

Academic Editor

PLOS ONE

Journal Requirements:

2. Thank you for stating the following in the Competing Interests/Financial Disclosure section:

JL is a now a fulltime employee of AstraZeneca with an honorary contract at Lancaster University. This research was completed prior to her employment at Astra Zeneca and they had no role in research. JL reported receiving personal fees for advisory boards from Novo Nordisk. This does not alter our adherence to PLOS ONE policies on sharing data and materials.

The remaining authors have no conflicts to declare.

We note that one or more of the authors are employed by a commercial company: AstraZeneca and Novo Nordisk

3. We noted in your submission details that a portion of your manuscript may have been presented or published elsewhere. Please clarify whether this publication was peer-reviewed and formally published. If this work was previously peer-reviewed and published, in the cover letter please provide the reason that this work does not constitute dual publication and should be included in the current manuscript.

5. Please remove all personal information, ensure that the data shared are in accordance with participant consent, and re-upload a fully anonymized data set.

Reviewers' comments:

Reviewer's Responses to Questions

**Comments to the Author**

1. Is the manuscript technically sound, and do the data support the conclusions?

Reviewer #1: Yes

Reviewer #2: Yes

2. Has the statistical analysis been performed appropriately and rigorously?

Reviewer #1: N/A

Reviewer #2: N/A

3. Have the authors made all data underlying the findings in their manuscript fully available?

Reviewer #1: Yes

Reviewer #2: Yes

4. Is the manuscript presented in an intelligible fashion and written in standard English?

Reviewer #1: No

Reviewer #2: No

Reviewer #1: I wish to thank the authors for providing the opportunity for me to review their article titled “Differences and commonalities in barriers and facilitators experienced by participants enrolled in an online behavioral weight management program: a qualitative comparison” aimed to understand how factors external to the program (i.e. intrapersonal, interpersonal, environment, and contextual) could influence weight loss during participation using the socio-ecological model framework to synthesise and discuss their findings. The authors found several common barriers and facilitators between ≥5% and <5% weight loss groups as well as some factors differing between them. The topic remains timely and the findings have important relevance. The methodology is well-grounded and appropriate for the proposed objectives. However, before the article can be considered for acceptance, certain aspects—particularly in the Methods section—require revision. The suggestions provided below are intended to contribute to the improvement of the manuscript and enhance its scientific value.

Key clarification

Methods: the authors interviewed the participants 4-8 weeks into the online programme. The categorisation into weight loss groups was added afterwards; but when was weight loss determined? This may be important in explaining the factors or their interactions. For example, an individual with a comorbidity or lifestyle disease may be more driven to achieve a weight goal. Also, an individual who has to achieve a mass media presence may be more driven.

Minor

1. Abstract

a. Kindly include the rationale for using the socio-ecological model in the Abstract.

2. Introduction

a. Should be better organized and paragraphs restructured to present the arguments more coherently. For example, a clearer background and contexts is needed on obesity and its health consequences, role and impact of weight management programmes (WMPs), including BWMPs as well as evidence of success of WMPs compared to BWMPs, which the authors alluded to. Then, the theoretical framework, SEM, need to be clearer; perhaps by clearly identifying the multi-level factors influencing participation in WMPs. For example, in lines 69-80, the example of Boehmer et al given in lines 96-100 could make what constitutes “changes in environment” clearer. Similarly, “changes to social interactions” should be clarified. Also, the phrase “individuals who achieve a ≥5% weight loss” need better emphasis on the significance of that measure and possibly the time frame to achieving it. Although the authors addressed this in methods, I consider it is more appropriate in Introduction where the cutoff was introduced.

b. The aim and objectives are clear.

c. Also the authors stated as follows “Additionally, data collection typically occurs at the program’s conclusion or during a follow-up phase, potentially overlooking essential nuances and meaningful barriers and facilitators experienced during participation.” This does not seem to have been addressed as a finding of this study. I consider that the aspect should be deleted from the Introduction.

3. Methods: Although this section mostly covers the necessary details, I somehow found some part of the section challenging. I believe the current sequencing of information in some parts may be confusing to the readers. Therefore, reorganising the content to reflect the heading or subheadings is important.

a. For example, it is unclear whether the wider mixed-methods project is the Second Nature online BWMP or that a mixed-methods study was conducted to evaluate the Second Nature online BWMP. Please make this clear under Study design and mention the overall study and its objectives to provide context for the qualitative component within the mixed methods design. Although I am uncertain how the Second Nature online BWMP programme fit with the mixed-methods project, I believe it should also be reported under the Study design. I suggest that the authors move ethical consideration details to a separate section because it is reported here and also partly duplicated in Recruitment (I seem to consider “Participants who completed the interview received £20 in form of a shopping voucher as compensation for their time and contribution to the research” an ethical issue than a Recruitment issue); Data collection (“Consent from participants was obtained either through email correspondence or recorded at the start of each interview. ……..audio files were uploaded and stored securely”).

b. The Recruitment section could be changed to “Participant selection and sampling” or a similar broader heading. Although the section has relevant information, it requires more regarding what was the basis for the sample size of 40-50? Is this based on a similar previous study or arbitrary? What type of sampling technique was used, convenient, purposeful, snowballing? Also, some details here would be more appropriate in other sections. For example, I consider “Interviews were conducted during weeks 4 to 8 of the 12-week program.” to be more appropriate in Data collection section where it is already; therefore, please delete. Also “ To prevent bias, coding was completed before the research team accessed participants' weight loss outcomes. Once coding was finalised, each participant's weight loss status (i.e., whether they achieved a 5% reduction) was added to their transcript to enable comparative analysis.” may be more appropriate in data analysis, where it is already, or in strength and limitations.

4. Results/discussions/conclusions:

The presentation of facilitators and barriers is clear and well-organized, providing a balanced overview of participant experiences. The authors added relevant direct participant quotations which strengthens authenticity and insight into the lived experience. The discussion is effective in integrating the findings with existing literature. Conclusions are appropriately drawn, emphasizing practical implications for program management and future research. Consider briefly clarifying about the influence of other unexplored issues e.g. existing health conditions or comorbidities, around how facilitators and barriers may interact or differ by subgroups for added depth.

The article will require editing and proofreading to enhance grammar, punctuation issues, consistency and flow.

Please see a few examples below

Abstract:

Correct punctuations errors: line 18

Grammar: lines 27-29;

Introduction

Incomplete: lines 78-81

Methods

Grammar: lines 244-246

Results;

Lacking clarity/grammar: lines 265-266

Conclusion:

Grammar: lines 700-702

Repetitions

E.g.: lines 187 (Data collection) and lines 255-256 (Results)

Consistency: Behavioral change vs behavior change

Tables should themes be fully capitalised vs not capitalised? E.g. Group Relations; Physiological response

Reviewer #2: Introduction:

The introduction provides a good background and contextualization of Behavioral Weight Management Programs in general. To further enrich the context, you may consider adding the following information:

• What is the average duration of such programs, or what is the minimum length of participation considered as completion?

• What are the exit criteria for these programs?

Results:

(Line 262) — Please justify describing only the female respondents in the descriptive table. It would be helpful to clarify whether this focus was based on sample composition, analytic relevance, or another reason.

Limitations:

(Lines 686–691) — The points raised in this section appear to align more closely with recall bias and the absence of data triangulation. You may wish to reframe them accordingly.

Minor Corrections:

• Please review the placement of the comma on line 18.

• Lines 80–81: Please check and revise these sentences for grammatical accuracy and clarity.

• Lines 92–95: Please review and correct these sentences to ensure coherence and precision.

**Do you want your identity to be public for this peer review?** For information about this choice, including consent withdrawal, please see our Privacy Policy

Reviewer #1: No

Reviewer #2: No

---

## [Author Response · Author response to Decision Letter 1]

19 Nov 2025

Dear Editor and reviewers,

Thank you for your time to review this work and for the constructive comments. I have reviewed and updated the manuscript based on these suggestions and feel the manuscript is much improved. All changes have been tracked. I have addressed the editor comments first then the reviewer. I hope this layout is acceptable – I have also uploaded a tracked and clean version of the document. Please find details of this below. I have added this as file "response to reviewers" as part of the submission.

Thank you again.

Yours Sincerely,

Dr Meigan Thomson

Editor/Journal Requirements Response

Please ensure that your manuscript meets PLOS ONE's style requirements, including those for file naming. I have reviewed the supplied style templates and updated the formatting to match these.

Please provide an amended Funding Statement declaring this commercial affiliation, as well as a statement regarding the Role of Funders in your study.

Please also provide an updated Competing Interests Statement declaring this commercial affiliation along with any other relevant declarations relating to employment, consultancy, patents, products in development, or marketed products, etc.

As far as I can tell this was already clarified in these sections. JL at the time of the research was employed by Lancaster University but did receive personal fees from Novo Nordisk for consulting on advisory boards. JL now works full time with astra Zeneca (post-study). Neither of these had any influence over the data collection or analysis. Please let me know if I have missed anything

We noted in your submission details that a portion of your manuscript may have been presented or published elsewhere. Please clarify whether this publication was peer-reviewed and formally published. If this work was previously peer-reviewed and published, in the cover letter please provide the reason that this work does not constitute dual publication and should be included in the current manuscript.

This work was assessed as part of a PhD thesis. Work from this thesis regarding the impact of COVID has been published in Clinical Obesity. The content of this article has only been put forward for publication to PLOS ONE

We note that you have indicated that there are restrictions to data sharing for this study. For studies involving human research participant data or other sensitive data, we encourage authors to share de-identified or anonymized data. Please update your Data Availability statement in the submission form accordingly.

This statement already states that as part of the consent participants only agreed for the data to be shared for academic research and teaching hence the restriction. I am unclear what else I need to add.

Please remove all personal information, ensure that the data shared are in accordance with participant consent, and re-upload a fully anonymized data set.

The data in the article are de-identified. I am unclear what is needed.

Thank you, I have reviewed the reference list and to the best of my knowledge it is correct.

Reviewer 1 comments Response

Abstract -

a. Kindly include the rationale for using the socio-ecological model in the Abstract. I have revised the abstract to meet the 300 word limit and be clearer on why the social ecological approach is suited

Introduction

a. Should be better organized and paragraphs restructured to present the arguments more coherently. For example, a clearer background and contexts is needed on obesity and its health consequences, role and impact of weight management programmes (WMPs), including BWMPs as well as evidence of success of WMPs compared to BWMPs, which the authors alluded to. Then, the theoretical framework, SEM, need to be clearer; perhaps by clearly identifying the multi-level factors influencing participation in WMPs. For example, in lines 69-80, the example of Boehmer et al given in lines 96-100 could make what constitutes “changes in environment” clearer. Similarly, “changes to social interactions” should be clarified. Also, the phrase “individuals who achieve a ≥5% weight loss” need better emphasis on the significance of that measure and possibly the time frame to achieving it. Although the authors addressed this in methods, I consider it is more appropriate in Introduction where the cutoff was introduced.

Thank you. I have updated the section and added in your suggestions regarding additional background on health, BWMPs and the SEM. I hope this strengthens the introduction and improves clarity.

Introduction

c. Also the authors stated as follows “Additionally, data collection typically occurs at the program’s conclusion or during a follow-up phase, potentially overlooking essential nuances and meaningful barriers and facilitators experienced during participation.” This does not seem to have been addressed as a finding of this study. I consider that the aspect should be deleted from the Introduction.

Thank you. I agree this is not highlighted in the discussion section. I think this is a really important unique aspect of the study so I have added this at the start of the discussion: Importantly, it captures these factors during program participation rather than relying on follow-up data collection, which can be affected by hindsight bias. This approach strengthens the study by offering real-time insights into the challenges participants face and a more nuanced understanding of the factors influencing weight management as they occur.

Methods

3. Methods: Although this section mostly covers the necessary details, I somehow found some part of the section challenging. I believe the current sequencing of information in some parts may be confusing to the readers. Therefore, reorganising the content to reflect the heading or subheadings is important.

a. For example, it is unclear whether the wider mixed-methods project is the Second Nature online BWMP or that a mixed-methods study was conducted to evaluate the Second Nature online BWMP. Please make this clear under Study design and mention the overall study and its objectives to provide context for the qualitative component within the mixed methods design. Although I am uncertain how the Second Nature online BWMP programme fit with the mixed-methods project, I believe it should also be reported under the Study design. I suggest that the authors move ethical consideration details to a separate section because it is reported here and also partly duplicated in Recruitment (I seem to consider “Participants who completed the interview received £20 in form of a shopping voucher as compensation for their time and contribution to the research” an ethical issue than a Recruitment issue); Data collection (“Consent from participants was obtained either through email correspondence or recorded at the start of each interview. ……..audio files were uploaded and stored securely”).

b. The Recruitment section could be changed to “Participant selection and sampling” or a similar broader heading. Although the section has relevant information, it requires more regarding what was the basis for the sample size of 40-50? Is this based on a similar previous study or arbitrary? What type of sampling technique was used, convenient, purposeful, snowballing? Also, some details here would be more appropriate in other sections. For example, I consider “Interviews were conducted during weeks 4 to 8 of the 12-week program.” to be more appropriate in Data collection section where it is already; therefore, please delete. Also “ To prevent bias, coding was completed before the research team accessed participants' weight loss outcomes. Once coding was finalised, each participant's weight loss status (i.e., whether they achieved a 5% reduction) was added to their transcript to enable comparative analysis.” may be more appropriate in data analysis, where it is already, or in strength and limitations. Thank you.

I have edited and restructured a bit which hopefully makes the section clearer.

I have updated the details on the wider study and added a Ethical considerations subsection above the reflexivity section. I have kept the information about incentives in the same place as this is detailing the process. The sampling strategy is described in the Recruitment section (i.e., participants who signed up to second nature within 2 weeks of the study period were sent an invitation email, those who were interested contacted the research to participate. This is a cross between purposeful and convenience (i.e., only inviting new participants of the program from the paid arm – but inviting them all)

Regarding the target sample size, success rates in these programs vary so widely. Since we were blind until after coding the data and the purpose of the interviews was to compare those who did and those who did not reach a target of 5% we set a high target with the hopes of having good representation of both groups. I have updated the section as follows: Since it was uncertain how many participants would reach a ≥5% weight lost, a target sample size of 40–50 participants was set to ensure sufficient representation in both groups—those who met the 5% target and those who did not. Previous research indicates those meeting these targets are variable but typically just under half of participants will reach this level of weight loss (10), so the sample size was chosen to account for variability and to capture experiences from both outcomes.

I agree with the notes on duplication/placement of materials and have removed these accordingly.

Discussion

Consider briefly clarifying about the influence of other unexplored issues e.g. existing health conditions or comorbidities, around how facilitators and barriers may interact or differ by subgroups for added depth.

Thank you for this consideration, I have updated the following paragraph: Moreover, the data mainly focuses on barriers and facilitators encountered by female participants with higher income and education levels. It is probable that male or non-binary individuals, as well as those with lower incomes orless formal education may experience distinct barriers and facilitators across various social ecological domains. It is also common for people living with obesity to experience a comorbid health condition, which could interact with how barriers and facilitators manifest. For example, a comorbid condition could affect motivation, access to resources, or the perceived cost-benefit of behavior change (74). Additionally, examining obstacles related to resource availability and environmental infrastructure could provide valuable insights into the challenges faced by individuals in deprived areas and communities (75).

Overall Comments

Abstract:

Correct punctuations errors: line 18

Grammar: lines 27-29;

Introduction

Incomplete: lines 78-81

Methods

Grammar: lines 244-246

Results;

Lacking clarity/grammar: lines 265-266

Conclusion:

Grammar: lines 700-702

Repetitions

E.g.: lines 187 (Data collection) and lines 255-256 (Results)

Consistency: Behavioral change vs behavior change

Tables should themes be fully capitalised vs not capitalised? E.g. Group Relations; Physiological response Thank you for these observations. I have made the following corrections.

Line 18 - replacement of the comma

Line 27-29 - rephrased to: Both groups highlighted willpower, knowledge, social support, and perceived safety in their local area as key facilitators, noting that their absence acted as barriers. The presence of food at home and the workplace was additionally reported as a barrier due to temptation.

Lines 78-81 – I have rephrased this section: Given that participation in BWMPs is integrated into individuals' daily lives, it's essential to consider how various social-ecological factors—ranging from personal to environmental—shape the adoption of weight-related behavior changes.

Lines 244-246 – I have revised this section and surrounding sentences as these lines referred to different sections in the clean/tracked versions so I was unclear which section you were referring to: These representatives provided feedback on the clarity of the questions and made suggestions for improvements. They recommended including questions on social norms and support within the household and workplace regarding dietary behaviors, as well as more detailed inquiries about their weight loss history. These suggestions were incorporated into the interview schedule. Furthermore, to reduce biases in data coding and interpretation, 10% of the transcripts were independently coded by SAS and AM. In addition, MT remained blinded to participant outcomes during coding of all of the transcripts allowing for an unbiased comparison of themes between the groups. These measures strengthened the study's credibility and reduced potential biases throughout the research process.

Lines 700-702. I have revised this section: These findings highlight areas that programs can target to better support participants, particularly those likely to achieve less than a 5% weight loss. Key areas for considerations for programs include helping participants manage reactions to barriers, addressing social conflict and support , and providing tailored advice to where the program is situated

Repetitions – I have removed the duration of the interviews from the results section

Consistency – I’ve updated table 2 themes so they are capitalised. Behavior change vs behavioral change are interchangeable terms

Reviewer 2 Comments Response

• What is the average duration of such programs, or what is the minimum length of participation considered as completion?

Thank you for this comment. I have not added this into the introduction as the duration of these programs vary so widely. Average duration of such programs vary but a typical goal in weight management is to lose 5% of your baseline weight by 12 weeks. Second Nature is a 12 week program (described in the methods section under recruitment)

• What are the exit criteria for these programs?

Thank you for this comment. I have not added this into the introduction because programs do not typically have exit criteria. A few programs in the US provided via the workplace have targets that employees need to reach but this isn’t the norm. There is no exit criteria for Second Nature. Participants from the UK are either given a 12 week block from their GP or pay for a 12 week block (the participants in this study were all paying). Once the 12 weeks ends they can chose to restart the program or not pay for another block. The material covered in each block is the same and the participants retain access to the materials and recipes. The other way they may exit the program is by dropping out (i.e., stop engaging/using the program)

(Line 262) — Please justify describing only the female respondents in the descriptive table. It would be helpful to clarify whether this focus was based on sample composition, analytic relevance, or another reason.

Thank you for your query.

The text and table do not refer exclusively to female participants. As indicated in the gender section, there were 48 participants in total (as shown in the column heading), of whom 40 were female and 8 were male. The table has been left unchanged, as this is a standard approach to reporting demographic data.

Both the table and the results section present data from the entire sample. All participants identified as either male or female, which is why no further breakdown of gender identity was provided.

Limitations:

(Lines 686–691) — The points raised in this section appear to align more closely with recall bias and the absence of data triangulation. You may wish to reframe them accordingly.

Thank you, this po

---

## [Decision Letter · Decision Letter 1]

7 Dec 2025

Differences and commonalities in barriers and facilitators experienced by participants enrolled in an online behavioral weight management program: a qualitative comparison.

PONE-D-25-30791R1

Dear Dr. Thomson,

We’re pleased to inform you that your manuscript has been judged scientifically suitable for publication and will be formally accepted for publication once it meets all outstanding technical requirements.

Kind regards,

Taiwo Opeyemi Aremu, MD, MPH, PhD

Academic Editor

PLOS One

Additional Editor Comments (optional):

Reviewers' comments:

Reviewer's Responses to Questions

**Comments to the Author**

Reviewer #1: All comments have been addressed

Reviewer #2: All comments have been addressed

2. Is the manuscript technically sound, and do the data support the conclusions?

Reviewer #1: Yes

Reviewer #2: (No Response)

3. Has the statistical analysis been performed appropriately and rigorously?

Reviewer #1: N/A

Reviewer #2: (No Response)

4. Have the authors made all data underlying the findings in their manuscript fully available?

Reviewer #1: Yes

Reviewer #2: (No Response)

5. Is the manuscript presented in an intelligible fashion and written in standard English?

Reviewer #1: Yes

Reviewer #2: (No Response)

Reviewer #1: The author has addressed all my concerns. I am happy for the article to be accepted for publication in Plos One

Reviewer #2: (No Response)

**Do you want your identity to be public for this peer review?** For information about this choice, including consent withdrawal, please see our Privacy Policy

Reviewer #1: No

Reviewer #2: No

---

## [Editor Report · Acceptance letter]

PONE-D-25-30791R1

PLOS One

Dear Dr. Thomson,

I'm pleased to inform you that your manuscript has been deemed suitable for publication in PLOS One. Congratulations! Your manuscript is now being handed over to our production team.

Kind regards,

on behalf of

Dr. Taiwo Opeyemi Aremu

Academic Editor

PLOS One